## [Peer Review File · Nature Communications]

Efficient multimode Wigner tomographyREVIEWER COMMENTS

Reviewer #1 (Remarks to the Author):

He et al. introduce a method for reconstructing the density matrix of a photonic multi-mode system in Fock space, using methods derived from Wigner tomography (displacements and parity detection). They analyze the sample complexity of the method and show an experimental implementation with a cQED system preparing a W-state in up to 4 modes. The experimental cost scales polynomially in the number of modes due to an imposed upper bound on photon number. The results are interesting and novel and generally technically correct. However, I have some issues with the claim of scalability/efficiency of the method, and consequently its scope of application, which the authors should address. And some other comments which are listed below.

Detailed comments:

1) The presentation of the sub-exponential scaling property is somewhat overselling it in my opinion: First of all it is accomplished by keeping the photon number fixed when increasing the mode number, in which case it is intuitively clear that the scaling is poly in M (size of the relevant Hilbert space).

Also, the emphasis on polynomial scaling suggests that the method is truly scalable, which it is not as the polynomial in M is quite high ($M^{(4N)}$ times a log term, according to equation 29 in the supplement).

The authors only exclusively consider W-state, raising the question, whether the method is at all feasibly extendible any states with higher photon number (which is claimed in the last sentence of the summary paragraph).

Some concrete points related to this (may be partly redundant):

- Abstract: "traditional tomography scales exponentially" -> State explicitly exponentially in what?

- Introduction: "scales exponentially with the size of the Hilbert space" is incorrect in my opinion.

- Second sentence of the introduction: I find the statement about exponentially greater complexity of quantum systems compared to classical ones imprecise/misleading. In the case of state tomography discussed here, one should compare reconstructing a quantum

state (Wigner function) of a system to reconstructing the probability distribution of a classical object over the same phase space. I don't see why the quantum one would be exponentially more complex. For a multi-mode system the e.g. number of points needed to represent the distribution would also scale exponentially in the mode number. Here the formulation should be made more precise.

- Comparison between OLI and DEMESST in Fig. 1: The exponential vs. polynomial scaling is presented as the main result, however, it is quite obvious where the difference comes from: Imposing the constraint of a finite total photon number vs. not imposing that constraint. So the question is how justified such a comparison is if the two methods simply make different assumptions about the underlying state (in particular DEMESST makes additional assumption which can lead to bias if they are not fulfilled.)

- Also, the advantage in sampling cost is much less clear in the experimental data. In particular, it is not obvious that the advantage further increase when going from 3 to 4 modes. The polynomial scaling is not demonstrated experimentally.

- Beginning of the results section: The statement "DEMESST scales polynomially with mode number for states that have support in a polynomial subspace of DFE-efficient basis operators" almost seems like a tautology. Should one make clear that restricting the Hilbert space is one thing, but measuring only the density matrix elements within this subspace is not easy in optics and is what the presented approach accomplishes?

2) End of intro: It is stated that "extracting fidelity bounds for W-states in 4 modes has not been accomplished yet and advances the state of the art". I am skeptical about his claim.

Isn't this more or less what was done in

<https://www.science.org/doi/10.1126/science.1172260>?

3) I noted that parts of the supplement are identical to the supplement of Ref 15 (the authors' own previous work) Could you more clearly differentiate the presented results from the ones already contained in earlier work?

Also a clearer distinction from other work using Fock space representation for Wigner tomography (e.g. <https://link.springer.com/article/10.1140/epjd/e2020-100649-3>) would be desirable.

4) It is stated that the objective of the procedure in the end is fidelity estimation: This typically does not require full quantum state tomography. Could one simplify the method by

only estimating the coherences that appear in the expression for the fidelity?

5) Questions on motivation/argumentation in intro: The authors state that QST is a fundamental part of quantum information processing. I would say it is not a part of the processing but rather is needed for the certification of devices. Giving examples would strengthen the argument.

6) In Fig. 4, the trace seems to significantly exceed 1 in some cases, meaning that the reconstructed state is unphysical. Is this a problem for fidelity estimation, since fidelities above 1 cannot be excluded? Could the problem be solved by enforcing the constraint of obtaining a physical density matrix (i.e. like in maximum likelihood methods)?

Minor comments:

- In the main text, N is used as the local Hilbert space dimension, while in the supplement it is the maximal total photon number

- “Here, we implement DEMESST on a multimode cavity system with Wigner tomography.” Wigner tomography is only historically what lead to employed technique for determining the density matrix elements. For me this was misleading since in the end you are working in the Fock state picture all the time. This also affects the title: Isn't it rather Fock space tomography? (But I acknowledge that there may be reasons to call this Wigner tomography, as this might lead to the correct association for people in the specific field...)

- “For 2 modes, this forms a dual rail encoding,” Is this information relevant? Seems a bit out of context. Actually, the correct statement would be that in a dual rail encoding, this would be a qubit state that lies on the equator of the Bloch sphere, right?

Supplement:

- “The Kerr nonlinearities range from $-6 - 7$ kHz.” Is a bit unfortunate notation. Make this consistent? Is there a “to” missing?

- Regarding the sampling procedure: Sample averages are calculated over a quantity with oscillating phase. Is this problematic as it can lead to exponentially increasing variance (sign problem), and this to very slow convergence? Can one say something about this?

- “displacements (see supplementary information)” What does this refer to? We are already in the supplement...

- Supplementary Fig. 3: It seems that some density matrix elements involving two-photon

states are non-zero, while in the text I find “Consequently, for the density matrix reconstruction, the Hilbert space will be spanned by $|n\rangle: \sum(n_m) \leq 1$.” Does this mean in Fig. 3 the constraint was rather $\sum(n_m) \leq 2$?

- I don't understand the statement “In particular, the W2 method can be used in a similar manner to the DEMESST, where the fidelity estimation is performed with respect to multimode Fock state basis elements. Repeating for multiple elements can thus provide a reconstructed density matrix.” The fidelities with a Fock states

With kind regards,

Martin Gärttner

Reviewer #2 (Remarks to the Author):

Review for “Efficient multimode Wigner tomography” by K. He et al.

The manuscript introduces a multimode Wigner tomography method applicable to states that have support over a subspace that scales polynomially with the number of modes. The method is tested in a circuit QED setup using a transmon dispersively coupled to multiple modes of a single storage 3D cavity. M -modes W states are prepared in the storage cavity using a photon blockade technique previously demonstrated in a similar experimental setup, described in Ref. 15, which are then reconstructed using the technique introduced here. It is demonstrated that the total number of measurements needed scales only polynomially with the number of modes M . In contrast, a more standard tomography method requires a total number of measurements that scales exponentially with the number of modes. The two methods are compared in the experiment and the demonstration of the advantage of the introduced method is clear.

Overall, the manuscript is well-written, and the results and calculations are convincing. I find the results very interesting, and I believe they will be of interest to a wide community of researchers working on quantum information processing. Certifying these kinds of multipartite entangled states in an efficient way is indeed a challenge very relevant today.

Here are detailed comments and questions:

1. In the introduction, the work is compared to other approaches. It is mentioned that Ref. 21 uses ancillary modes and a known excitation number. In the present work, the transmon could be described as an ancilla, and the tomography scheme does take advantage of the known excitation number of the state. I'm therefore confused about whether that qualification of Ref. 21 would also apply to the present work. Especially, what does it mean "to obtain more full state information" in comparison to Ref. 21?

2. In the introduction, it reads "It only requires local operations if the subspace is spanned by a set of finite local operations acting on a DFE-efficient state [22]." I didn't understand what was said here. Could the authors clarify? Did you mean "a finite set" instead of "a set of finite"?

3. Just before equation (1), it is mentioned that the state is reconstructed without introducing bias. After reading the supplemental information, I think you are referring to the bias that would be introduced if some displacements are purposely not sampled to avoid some pathological behaviour. Yet the statement in the main text is a bit ambiguous as it is. Perhaps you could elaborate a bit further on what bias you are referring to.

4. In the caption of Figure 2, it is not clear what "it" is in "to project it onto the real axis".

5. On page 3, right column, first paragraph, it is mentioned that "For two modes, OLI performs better due to overhead required for the DEMESST approach (see supplementary information)". Reading the SI, it was not clear to me what is the overhead you are referring to.

6. On page 4, left column: "These pulses project one or more of the cavity modes to the transmon $|f\rangle$ level...". I get what you are trying to say, but I feel that "projecting" cavity modes to transmon states is not the right way of describing what you are doing. Consider perhaps rephrasing this.

7. A few lines later, again the word “project” is used two more times. Consider perhaps rephrasing these sentences.
8. I think that details are missing in the discussion about the simulations in the SI. How exactly do you simulate the Wigner tomography? Enough details should be given for the readers to be able to reproduce the calculations. It would be useful to have them explained in the SI.
9. In the simulated fidelities reported in Table 1, were the parameters used in the simulations obtained independently from the Wigner tomography experiments? Or some simulation parameters were fitted to match the experimental results? I think this should be commented.
10. What is the transmon readout error? Can you comment on how the readout error, transmon thermal excitations, transmon decay, and transmon dephasing influence the number of shots (10 in your experiment) needed per displacement vector?
11. On page 4, when discussing the number of distinct displacements in DEMESST, it is mentioned that the Total Measurement Number is divided by 10 and by the number of elements. What do you mean by “number of elements”?
12. In Section II of the SI, the range of the Kerr nonlinearities given might have typo. According to Table 2, they range from -6 KHz to +9 KHz.
13. In equation (8) in SI, it is not immediately obvious to me why or how the real part is taken. Is it a redundancy to communicate that, despite the complex phase, the quantity you take the real part of is real?
14. In equation (20) in the SI, what is $p_D(\vec{\alpha}_{\bar{S}})$?
15. In the paragraph just before Section VI of the SI, it says “(see supplementary

information)", but we are already there.

Cristóbal Lledó

REVIEWER COMMENTS

Reviewer #1 (Remarks to the Author):

He et al. introduce a method for reconstructing the density matrix of a photonic multi-mode system in Fock space, using methods derived from Wigner tomography (displacements and parity detection). They analyze the sample complexity of the method and show an experimental implementation with a cQED system preparing a W-state in up to 4 modes. The experimental cost scales polynomially in the number of modes due to an imposed upper bound on photon number. The results are interesting and novel and generally technically correct. However, I have some issues with the claim of scalability/efficiency of the method, and consequently its scope of application, which the authors should address. And some other comments which are listed below.

Dear Martin,

Thank you for your perusal of our manuscript and your insightful comments. We are glad you found our results interesting and novel. We agree with your points, and have made changes throughout the paper to address the issues presented by them. As a brief summary of those alterations, we have added details for clarity and toned back some of our language to avoid over-exaggerating our results throughout the main text. Also, we draw more distinctions between our work and previous ones, and emphasize that the objective of the DEMESST tomography method is state reconstruction. Overall, thank you again for helping us significantly improve the quality of our paper. Please find detailed responses to your comments below. Your original comments are in black, and our responses are in blue. Additionally, the changes to the text in the updated article files are marked in red.

Detailed comments:

1) The presentation of the sub-exponential scaling property is somewhat overselling it in my opinion: First of all it is accomplished by keeping the photon number fixed when increasing the mode number, in which case it is intuitively clear that the scaling is poly in M (size of the relevant Hilbert space).

We acknowledge these points about the source of the advantage and the intuitiveness of our protocol. Indeed, the dimension of the relevant Hilbert space

scales polynomially in M , which suggests that if arbitrary unitary operations can be executed in such a subspace, we should be able to recover the full state information with $\text{poly}(M)$ resources.

Besides the scaling, our approach has other unique merits as well. One of them, as we also describe when addressing Reviewer #2's first concern about the similarity between our work and Ref. 21, is that in principle, our approach requires only local operations (displacements, parity measurements, and vacuum state projections in each mode). Due to hardware limitations in our experimental demonstration, where we have only one transmon that couples with all the modes, we perform joint (generalized) parity and vacuum projections, which would be unnecessary if each mode was coupled to a separate transmon. Our tomography protocol would still be feasible when each mode is separated, i.e., there is no direct coupling among them, which would be useful for distributed quantum networks.

Another novelty in our protocol is the use of the vacuum state projection when estimating each element of the density matrix in the Fock basis. If we just sampled and measured to estimate each element based on the M -mode Wigner from the DFE method, we would still suffer from an exponential 2^M factor in sampling overhead (as shown in Eqn. (18) in the SI) to obtain reasonable statistical accuracy. To resolve this issue, we make use of the projection trick in Eqn. (19) in the SI to only focus on the relevant modes in our matrix element estimation. This provides an efficient way to get unbiased estimation of each element in the density matrix, which, in combination with the fact that there is a $\text{poly}(M)$ amount of them, gives the final efficiency claim of our protocol.

Nevertheless, we certainly do want to avoid overselling our results. We have increased the accuracy of the statements we make about them by emphasizing the conditions where DEMESST applies, and have made the following modifications:

- 1) in the abstract, specify that the sampling scales polynomially if the state can be expressed in a polynomial subspace
- 2) in the caption of Fig. 1(a), emphasize that we sample in polynomial subspaces
- 3) in the "Simulated performance of OLI and DEMESST" section, specified that

DEMESST scales polynomially when we have a fixed maximum photon number

In general, we have added more detailed qualifiers and reminders of the conditions where DEMESST applies throughout the paper, and have also made several corrections for clarity, as described in responses to later points.

Also, the emphasis on polynomial scaling suggests that the method is truly scalable, which it is not as the polynomial in M is quite high (M^{4N} times a log term, according to equation 29 in the supplement).

The goal of our work is to provide an unbiased way to perform tomography for a generic mixed state in a confined subspace with dimension $O(M^N)$, where the number of unknown variables for the density matrix already scales as $O(M^{2N})$. If we do not make any further assumptions about the structure of the state (for example, that the state has a low rank, where it is possible to use compressed sensing techniques to further reduce the sampling requirement), we have to take at least an $O(M^{2N})$ amount of data to achieve "information completeness." Because of this, we do not think the $O(M^{4N})$ scaling is a drawback of our protocol, as we do have a large amount of unknown variables to be estimated, and the slightly larger exponent—which still scales polynomially with $O(M^{2N})$ —arises from bounding the Frobenius norm matrix distance error of our reconstruction from the true result.

As a side note, we discuss the potential application of compressed sensing techniques in more detail in response to the later comment about the previous work by Botelho et al.

The authors only exclusively consider W-state, raising the question, whether the method is at all feasibly extendible any states with higher photon number (which is claimed in the last sentence of the summary paragraph).

This is a valid concern to have. There is always a fundamental limitation set by the number of unknown variables of the state and our prior knowledge of the state. If the dimension of the Hilbert space that the unknown state lies in grows exponentially in mode number M and we have no prior knowledge, then none of the tomography methods will be efficient.

On the other hand, when deriving the DEMESST method in the SI, we consider the state tomography task for a generic state with maximum total photon number N distributed among M modes, where $M > 2N$. We demonstrate a protocol that performs unbiased estimation of the density matrix, and prove that the sampling overhead is $\text{poly}(M)$ in the large M limit when N is bounded. Therefore, if these conditions are satisfied, the method should be applicable to larger N . In general, if we have an ansatz for the form of the state, or know that it has a low rank, we may save some sampling overhead. For DEMESST, it helps if the Hilbert space of interest consists of the following types of states: a "base state" on which one can easily perform DFE, and "nearby states" that come from a set of finite-party (independent of M) operations acting on a base state.

To make the summary more accurate, we have toned back the language and specified that "the DEMESST sampling method enables efficient reconstruction of certain large multiqubit or multimode states, and will be advantageous as the size of quantum hardware increases..."

Some concrete points related to this (may be partly redundant): - Abstract: "traditional tomography scales exponentially" -> State explicitly exponentially in what?

Apologies for the missing detail here, we have modified that sentence to now read: "When characterizing these states, traditional tomography scales exponentially with the number of modes in both computational and experimental measurement requirement, which becomes prohibitive as the system size increases."

- Introduction: "scales exponentially with the size of the Hilbert space" is incorrect in my opinion.

Thank you for pointing out this incorrect phrasing, the Hilbert space itself is what generally scales exponentially with the size of the system. We have modified these mentions (in the main text and in the caption for Figure 1) to now read: "scales exponentially with the size of the system" and "scales exponentially with the number of modes M ," respectively.

- Second sentence of the introduction: I find the statement about exponentially greater complexity of quantum systems compared to classical ones imprecise/misleading. In the case of state tomography discussed here, one should compare reconstructing a quantum state (Wigner function) of a system to reconstructing the probability distribution of a classical object over the same phase space. I don't see why the quantum one would be exponentially more complex. For a multi-mode system the e.g. number of points needed to represent the distribution would also scale exponentially in the mode number. Here the formulation should be made more precise.

This is a good point. As we do not want to distract from the main point of our paper, which is not directly related to classical systems, we have decided to remove this sentence.

- Comparison between OLI and DEMESST in Fig. 1: The exponential vs. polynomial scaling is presented as the main result, however, it is quite obvious where the difference comes from: Imposing the constraint of a finite total photon number vs. not imposing that constraint. So the question is how justified such a comparison is if the two methods simply make different assumptions about the underlying state (in particular DEMESST makes additional assumption which can lead to bias if they are not fulfilled.)

In terms of justifying the comparison, our goal is to compare different techniques for achieving full state reconstruction: our newly developed DEMESST method, and the OLI approach which has seen prevalent use otherwise.

OLI does require a defined finite total photon number to make processing tractable, but utilizes that condition in a different way, typically on each mode individually rather than across the entire system as a whole. Thus, as we have described in some of our responses to previous comments, applying constraints on the form of the Hilbert space, like we do in DEMESST, is necessary to reduce the number of parameters for a generic unknown mixed state to scale as $\text{poly}(M)$, which makes the full state efficient tomography task possible. Our DEMESST method also enjoys other features, such as in principle only needing single-mode operations. It is not necessarily the most efficient sampling method, but is one with unbiased estimation that was achievable on our hardware and has $\text{poly}(M)$ sampling overhead.

With regards to the bias, one advantageous feature of DEMESST is that it always provides unbiased estimation for the expectation value of the observable of interest. In our case, these are density matrix elements in the multimode Fock basis. Consequently, we can always perform a self-consistency check, like seeing if the trace of the matrix in the subspace we choose equals 1 in Fig. 4, to make sure the assumption (total photon number $N = 1$) is valid. In particular, if we find that the reconstructed density matrix in the assumed subspace has a high probability of having a trace less than 1, we would know that there exists nonzero population outside of that subspace. After detecting this, we could then adjust our assumption and measure the expectation values of other states (such as those with 2 photons or with population in other modes, in our case).

- Also, the advantage in sampling cost is much less clear in the experimental data. In particular, it is not obvious that the advantage further increase when going from 3 to 4 modes. The polynomial scaling is not demonstrated experimentally.

Admittedly, with what is abstractly three data points ($M = 2, 3, 4$), it may be difficult for us to claim polynomial scaling based purely on the experimental data. However, we have theoretical evidence for the behavior that we observe in the experiment, in the form of derivations and simulations detailed in the SI. Also, the advantage (at least in the form of Frobenius norm matrix distance) does seem to further increase when going from 3 to 4 modes, as indicated by the matrix distance ratios in the rightmost column of Table 1 that compare OLI with DEMESST. With the theoretical results in mind, and noting that DEMESST gains its advantage from sampling a polynomial subspace so that it makes sense intuitively that the sampling should scale polynomially with mode number as you have pointed out, we believe that the data is a sufficient demonstration of the expected polynomial scaling.

- Beginning of the results section: The statement “DEMESST scales polynomially with mode number for states that have support in a polynomial subspace of DFE-efficient basis operators” almost seems like a tautology. Should one make clear that restricting the Hilbert space is one thing, but measuring only the density matrix elements within this subspace is not easy in optics and is what the presented approach accomplishes?

Thank you for mentioning this, it is true that the statement is very intuitive, as you described in an earlier comment, so being able to measure only the density matrix elements within our target subspace is a nice accomplishment to highlight. Thus, we have added the sentence to the first paragraph of the Results section: "DEMESST allows us to individually sample density matrix elements in a subspace of interest."

2) End of intro: It is stated that “extracting fidelity bounds for W-states in 4 modes has not been accomplished yet and advances the state of the art”. I am skeptical about his claim. Isn’t this more or less what was done in <https://www.science.org/doi/10.1126/science.1172260>?

We acknowledge that Papp et al. (which we will refer to as [Pa09]) performed a similar experiment, where a W-state in 4 spatially distinct optical modes was prepared and somewhat characterized. However, in [Pa09], the authors focus on an “entanglement witness” measurement; that is, their main goal is to measure a single quantity and use that to claim that all four modes are entangled, with none of them separable from the others. In [Pa09], there is no report of the fidelity between the state they prepared experimentally and the desired state, and also no report for full state tomography.

On the other hand, in our work, we experimentally demonstrate full state tomography for a generic state with a maximum of one photon distributed in up to 4 modes (with the expected state being a 4-mode W state), and fully reconstruct the density matrix of the state. This is more demanding than the entanglement witness measurement. Then, as a byproduct, we calculate and plot the fidelity of the prepared W state based on the reconstructed density matrix obtained from our measurements.

To highlight the main achievement in our work, we have rephrased this sentence as follows: "With this approach, we measure the density matrices of W states prepared in up to 4 bosonic modes, which is beyond existing demonstrations and advances the state of the art."

Additionally, we have added a citation to the suggested reference [Pa09] and added it to our discussion of W states in the introduction, so that it now reads: "In particular, W states have unique multipartite entanglement—demonstrated

with entanglement witnesses [23], [24]—and protection against photon loss that gives them applications in quantum communication.

3) I noted that parts of the supplement are identical to the supplement of Ref 15 (the authors' own previous work) Could you more clearly differentiate the presented results from the ones already contained in earlier work?

This is true, we use the same device as in the previous work, although with some wiring changes. Additionally, we provide the system Hamiltonian in Eqn. (1) of the SI with updated experimental parameters in STable 1, and they are expectedly the same as or very similar to the previous work. Nevertheless, we thought these elements useful to include.

However, we agree that we should focus on the changes between works and not repeat most of the unnecessary details. To better highlight the changes between this work and our previous one, we have removed most of the repeated description in Sec. I of the SI, so that it now reads: "Like in [1], the multimode cavity device is heat sunk to an OFHC copper plate connected to the base stage of a Bluefors LD-400 dilution refrigerator (10-12 mK). A schematic of the cryogenic setup, control instrumentation, and device wiring is shown in SFig. 1. Compared to [1], there is an additional cavity drive line that directly couples to the multimode cavity, rather than going through the transmon. Furthermore, a JPA was added on the output to improve the readout fidelity and allow us to utilize lower readout powers, as described at the end of Sec. VI. The JPA was powered by an SRS source and controlled with one channel of a Quantum Machines OPX Controller. Controls are performed through the readout cavity or the direct cavity drive line, by driving at the qubit and storage mode frequencies."

Also, we have added the phrases "Like in [1]" and "(updated from [1])" to Sec. II of the SI to differentiate the results from this work vs. the previous one.

Also a clearer distinction from other work using Fock space representation for Wigner tomography (e.g. <https://link.springer.com/article/10.1140/epjd/e2020-100649-3>) would be desirable.

We acknowledge that we should differentiate our work from the one by Botelho

et. al. (which we will refer to as [Bo20]). Here, we highlight several main differences between our work and theirs.

In [Bo20], they consider the tomography task with "informationally incomplete data;" that is, the problem becomes a compressed sensing/matrix completion problem which is in favor of reconstructing a state with low rank. In our work, we mainly focus on the multimode setup. Due to our assumed subspace (whose dimension is "small" since it scales as $\text{poly}(M)$ rather than $\exp(M)$), we accumulate an "informationally complete" amount of data to suppress the statistical fluctuation. After choosing the subspace, we do not apply an additional "low rank" assumption on the state lying in it. We also provide a Hoeffding's bound on the estimated resource overhead (which again scales as $\text{poly}(M)$) given the reconstructed accuracy. In [Bo20] however, there is no clear analytical estimation of the amount of resources necessary to achieve a certain accuracy.

Another difference is the quantity we measure. In our work, we use (generalized) parity measurements to get (generalized) Wigner values for each sampled point in phase space. However, in [Bo20], they mainly assumed homodyne / heterodyne detection, and achieved a distribution related to the Wigner or Husimi Q function of points in specific quadratures/phase space, then used that to reconstruct the states.

We also note that even with compressed sensing techniques, one must still choose an appropriate basis for the state to avoid an exponential sampling overhead. In general, compressed sensing can have a different scaling than that of our DEMESST subspace assumption. Combining the two approaches and implementing compressed sensing (when having the information appropriate for making those additional assumptions) in a DEMESST subspace to obtain even greater efficiency could be of interest for future work, but we believe that it lies beyond the scope of our current one.

We have added a sentence summarizing this in the introduction, which reads: "Even with techniques like compressed sensing [14-16], the sampling and number of measurements required can still scale exponentially with the number of modes."

4) It is stated that the objective of the procedure in the end is fidelity estimation: This typically does not require full quantum state tomography. Could one simplify the method by only estimating the coherences that appear in the expression for the fidelity?

We apologize for being unclear about the objective of our procedure. The goal of our work is to fully reconstruct an unknown (mixed) state with a bounded total photon number across all the modes. The fidelity estimation is a byproduct of the final result, which is full state tomography and density matrix reconstruction. To clarify this, and also address one of your earlier points and one of Referee 2's comments, we have rewritten parts of the introduction to say "we instead...directly measure and reconstruct the density matrices of potentially mixed states...", and "With this approach, we measure the density matrices of W states..." We have also changed the statement "faster convergence to the final state fidelity" to "faster convergence to the final state" in the discussion of Fig. 3.

5) Questions on motivation/argumentation in intro: The authors state that QST is a fundamental part of quantum information processing. I would say it is not a part of the processing but rather is needed for the certification of devices. Giving examples would strengthen the argument.

This is a proper distinction to make. We have modified the first sentence of the introduction to now read: "Quantum state tomography (QST) is the process of determining the quantum state of a system, and is a crucial part of certifying operations and characterizing processes in quantum information science."

6) In Fig. 4, the trace seems to significantly exceed 1 in some cases, meaning that the reconstructed state is unphysical. Is this a problem for fidelity estimation, since fidelities above 1 cannot be excluded? Could the problem be solved by enforcing the constraint of obtaining a physical density matrix (i.e. like in maximum likelihood methods)?

This is a good observation. For the data shown in Fig. 3, we do impose physicality constraints (trace 1 and positive semidefiniteness) in both the DEMESST and OLI methods, using maximum likelihood as you have described. The goal of Fig. 4 is to show that with sufficient measurements, we can still converge to an

accurate density matrix reconstruction using the DEMESST method even without imposing those constraints. This can be useful as a check to see whether our prepared state has support outside our expected measured space, as we discuss in the main text.

Minor comments: - In the main text, N is used as the local Hilbert space dimension, while in the supplement it is the maximal total photon number

Thank you for catching this inconsistency, to avoid confusion with the supplement we have replaced the N 's in the main text with d 's.

- "Here, we implement DEMESST on a multimode cavity system with Wigner tomography." Wigner tomography is only historically what lead to employed technique for determining the density matrix elements. For me this was misleading since in the end you are working in the Fock state picture all the time. This also affects the title: Isn't it rather Fock space tomography? (But I acknowledge that there may be reasons to call this Wigner tomography, as this might lead to the correct association for people in the specific field. . .)

We acknowledge this potential terminological issue. We call our work "Wigner tomography" because we measure (generalized) Wigner function values at different points in phase space and use these values to reconstruct unknown states. It is true that to get a representation of the state, we choose Fock states, as they fit well into our "bounded total photon number" assumption, but this is solely a choice of representation. The key message we want to convey is that our states are reconstructed based on measurements of Wigner functions, and so we believe that referring to the measurements as Wigner tomography is still appropriate.

- "For 2 modes, this forms a dual rail encoding," Is this information relevant? Seems a bit out of context. Actually, the correct statement would be that in a dual rail encoding, this would be a qubit state that lies on the equator of the Bloch sphere, right?

This is a good point. Upon further consideration, we agree that this statement regarding dual rail is not relevant, so we have removed that part of the sentence.

It now reads: "For 2 modes, an ideal W state is given by..."

Supplement: - "The Kerr nonlinearities range from -6 - 7 kHz." Is a bit unfortunate notation. Make this consistent? Is there a "to" missing?

Thank you for this observation, we have corrected the text to the correct range that was provided in the table and switched to using the word "to" to indicate range (rather than the dash), so it now reads "from -6 to +9 kHz."

- Regarding the sampling procedure: Sample averages are calculated over a quantity with oscillating phase. Is this problematic as it can lead to exponentially increasing variance (sign problem), and this to very slow convergence? Can one say something about this?

We acknowledge this issue, known as the sign problem. In this work, we fix the photon number N and focus on the scaling with mode number M , thereby avoiding a variance that increases exponentially with N . In more detail, since our sampling distribution function is proportional to the absolute value of the Wigner function, we need to introduce a normalization factor Z (defined in Eqn. (12) in the SI), which contributes to the sampling overhead (Eqn. (14) & Eqn. (22) in the SI). The value of Z in general depends on the basis operator O (defined in Eqn. (15) in the SI).

For example, in the case of a single mode, using numerical integration, we indeed find that Z will increase as the corresponding O is associated with more photons. In this calculation, we go up to $n \leq 40$, as there may be numerical precision issues beyond that. These issues are due to the integrand, which is proportional to the absolute value of the generalized Laguerre polynomial and is highly oscillatory there. Denote $Z_{n,n}$ and $Z_{n,m}^{R(I)}$ as the normalization factors of $O_{n,n}$ and $O_{n,m}^{R(I)}$ respectively, where R or I indicate the forms of the operators like in Eqn. (15) of the SI. We see that $Z_{n_1,n_1} < Z_{n_2,n_2}$ when $n_1 < n_2$. However, $Z_{n,n}$ grows slowly as n increases, rather than exponentially, as $(Z_{n+1,n+1} - Z_{n,n}) < (Z_{n,n} - Z_{n-1,n-1})$ for $n \leq 40$. Also $Z_{n_1,n_2}^{R(I)} < Z_{\max(n_1,n_2),\max(n_1,n_2)}$. Therefore, the oscillatory integrand does not lead to an exponentially growing sampling overhead as the photon number increases.

The multimode case is a generalization of the single-mode case. For example, the Wigner function for the diagonal basis elements $O_{\vec{n},\vec{n}}$ can be written as the product of each single-mode Wigner: $\langle \vec{n} | D(\vec{\alpha}) e^{i\pi \sum_m a_m^\dagger a_m} D^\dagger(\vec{\alpha}) | \vec{n} \rangle = \prod_m \langle n_m | D(\alpha_m) e^{i\pi a_m^\dagger a_m} D^\dagger(\alpha_m) | n_m \rangle$, so that $Z_{\vec{n},\vec{n}} = \prod_m Z_{n_m,n_m}$. Similarly, for the off-diagonal elements $O_{\vec{n}_1,\vec{n}_2}^{R(I)}$, the Wigner function can be written as the

product of a radial part and an angular part $W(\vec{\alpha}) = R(\{|\alpha_m|\}) \cdot \Psi(\{\psi_m\})$ (here $\alpha_m := |\alpha_m|e^{i\psi_m}$). Only the radial part $R(\{|\alpha_m|\})$ contributes non-trivially as photon number increases, and it is related to the single-mode results as $R(\{|\alpha_m|\}) = \prod_m R_{n_{1m}, n_{2m}}(|\alpha_m|)$, whose behavior as photon number grows was discussed previously. To get exponential N -dependence of sampling overhead, one needs to consider basis operators where N photons are distributed across more modes. For example, if we choose $|\vec{n}\rangle = |11\dots 1\rangle$, then $Z_{\vec{n}, \vec{n}} = (Z_{1,1})^N$, which has exponential dependence on N , as expected.

However, there is no need to consider the discussion above in our problem setup, as we focus on the large mode number M regime with a bounded total photon number N and estimate the resource overhead scaling over M . With the projection method we introduce (see Eqn. (19) & Eqn. (22) in the SI), the sampling overhead of each basis operator only depends on its nontrivial part, $C_{\vec{s}}Z_{O_{\vec{s}}}$, which is independent of M when $M > 2N$.

We have added a sentence addressing this near the end of Sec. V of the SI: "We note that in the above, we consider a fixed N and focus on the scaling over M . Thus, since the subject of our work is the protocol's dependence on M , we do not encounter or address the sampling overhead's exponential dependence on N , which is also known as the sign problem."

- "displacements (see supplementary information)" What does this refer to? We are already in the supplement...

Thank you for catching that mistake, we have removed the "see supplementary information" phrase.

- Supplementary Fig. 3: It seems that some density matrix elements involving two-photon states are non-zero, while in the text I find "Consequently, for the density matrix reconstruction, the Hilbert space will be spanned by $|n\rangle$: $\text{sum}(n_m) \leq 1$." Does this mean in Fig. 3 the constraint was rather $\text{sum}(n_m) \leq 2$?

Apologies for the confusion, the statement in the text in Sec. VI is correct in that context, as a W state should live entirely in that space. However, for SFig. 3, we did apply slightly different constraints, to allow us to directly measure

some state preparation errors. For DEMESST (left column in SFig. 3), the constraint was the correction you wrote above, while for OLI, we included the full 0 and 1 photon space tensored for each mode. To help clarify this point, we have added the following to Sec. VII of the SI around the discussion of SFig. 3: "Additionally, to directly observe state preparation errors, for DEMESST we measure the multimode Fock basis elements with up to 2 photons, and for OLI we measure the full tensored $\{|0\rangle, |1\rangle\}^{\otimes M}$ space."

- I don't understand the statement "In particular, the W^2 method can be used in a similar manner to the DEMESST, where the fidelity estimation is performed with respect to multimode Fock state basis elements. Repeating for multiple elements can thus provide a reconstructed density matrix." The fidelities with a Fock states

It seems that the end of this comment has been cut off, but we still want to address this and explain the statement we made here. The W^2 method (see Ref. [5, 6] in the SI) is another choice of sampling distribution function (see Eqn. (8) in SI) for direct fidelity estimation where $p(\vec{\alpha}) \propto |W_O(\vec{\alpha})|^2$, in contrast to DEMESST, where $p(\vec{\alpha}) \propto |W_O(\vec{\alpha})|$. As a result, we can also use the W^2 method to estimate the expectation value $\text{Tr}[\rho O]$ via Eqn. (8) in the SI. If we go through all the basis operators (given by Eqn. (15) in the SI) in the subspace we focus on, we will get every density matrix element and therefore a reconstructed density matrix, in a similar manner to DEMESST.

In Sec. IX of the SI, however, instead of choosing O as the basis operators given by SI Eqn. (15), we choose $O = |W\rangle\langle W|$ as the density matrix of a pure W -state. We obtain $|W\rangle$ experimentally from the best fits using DEMESST and OLI explained in Sec. VII of the SI. In this way, we use the W^2 method to further confirm the W state fidelities obtained using DEMESST and OLI.

To help clarify this, we have modified the discussion of the W^2 method in the SI to the following: "This approach allows for direct fidelity estimation of a prepared state compared with an ideal state. The W^2 method can also be used in a similar manner to DEMESST, where the fidelity estimation of a prepared state is performed with respect to an element of the multimode Fock state basis

to determine the value of that matrix element. Repeating this for multiple different basis elements can thus provide a reconstructed density matrix."

With kind regards,

Martin Gärttner

Reviewer #2 (Remarks to the Author):

Review for “Efficient multimode Wigner tomography” by K. He et al.

The manuscript introduces a multimode Wigner tomography method applicable to states that have support over a subspace that scales polynomially with the number of modes. The method is tested in a circuit QED setup using a transmon dispersively coupled to multiple modes of a single storage 3D cavity. M -modes W states are prepared in the storage cavity using a photon blockade technique previously demonstrated in a similar experimental setup, described in Ref. 15, which are then reconstructed using the technique introduced here. It is demonstrated that the total number of measurements needed scales only polynomially with the number of modes M . In contrast, a more standard tomography method requires a total number of measurements that scales exponentially with the number of modes. The two methods are compared in the experiment and the demonstration of the advantage of the introduced method is clear.

Overall, the manuscript is well-written, and the results and calculations are convincing. I find the results very interesting, and I believe they will be of interest to a wide community of researchers working on quantum information processing. Certifying these kinds of multipartite entangled states in an efficient way is indeed a challenge very relevant today.

Dear Cristóbal,

Thank you for your careful reading of our manuscript and your insightful comments. We are happy you found our results interesting and relevant. We agree with your points, and have made changes throughout the paper to address the issues raised by them. As a summary, we have clarified our language throughout the main text. Also, we have added comprehensive details regarding our simulations and transmon readout error to the supplementary information. Overall, thank you again for helping us greatly improve the quality of our paper. Please find specific responses to your comments below. Your original comments are in black, and our responses are in blue. Additionally, the changes to the text in the updated article files are marked in red.

Here are detailed comments and questions:

1. In the introduction, the work is compared to other approaches. It is mentioned that Ref. 21 uses ancillary modes and a known excitation number. In the present work, the

transmon could be described as an ancilla, and the tomography scheme does take advantage of the known excitation number of the state. I'm therefore confused about whether that qualification of Ref. 21 would also apply to the present work. Especially, what does it mean "to obtain more full state information" in comparison to Ref. 21?

Thank you for indicating this source of confusion. We acknowledge that there are some underlying connections between our work and Ref. 21. We think one of the main differences is that our sampling protocol, in principle, does not require entangling operations between modes for state characterization. Specifically, we only need displacement operations, parity measurements, and vacuum state projections for each individual mode. We did not demonstrate the latter two in experiment, because we only have one transmon that dispersively couples to all modes, rather than having each mode individually couple with its own transmon. In contrast, the protocol in Ref. 21 requires entangling operations (beam-splitters) between modes, like the $U_{\text{bs}}^{(1,2)}$ gate in their Fig. 3(a). Therefore, we believe our sampling protocol will be more feasible in situations where the modes are distributed, i.e., there is no direct coupling between all pairs of them.

We apologize for our misleading phrasing. We mean to say that our method aims to reconstruct the full density matrix for a generic mixed state, which is the same as what they do in Ref. 21.

To address this, we have rephrased the sentence as: "In our case, to obtain full state information, we instead use a technique that in principle only requires local operations to directly measure and reconstruct the density matrices of potentially mixed states confined in a subspace of interest." We have also added to the summary at the end of the paper: "This approach can in principle operate without coupling gates between modes, such as when each mode has its own transmon for performing parity measurements, ..."

2. In the introduction, it reads "It only requires local operations if the subspace is spanned by a set of finite local operations acting on a DFE-efficient state [22]." I didn't understand what was said here. Could the authors clarify? Did you mean "a finite set" instead of "a set of finite"?

We agree that this statement was difficult to interpret. We mean to say that DEMESST applies in subspaces that are obtained by acting finite local operations on a DFE-efficient state. We have made the following change for clarity: "When the subspace is spanned by a set of states obtained by acting finite local operations on a DFE-efficient state, the operations required to implement DEMESST are local as well [27]." (this is now Ref. 27 instead of 22, as we have added several references related to comments from Reviewer 1)

3. Just before equation (1), it is mentioned that the state is reconstructed without introducing bias. After reading the supplemental information, I think you are referring to the bias that would be introduced if some displacements are purposely not sampled to avoid some pathological behaviour. Yet the statement in the main text is a bit ambiguous as it is. Perhaps you could elaborate a bit further on what bias you are referring to.

Apologies for the ambiguity, the "bias" means statistical bias. To estimate the value of density matrix elements in the Fock basis, we sample according to the distribution $p_D(\vec{\alpha}) \propto |\tilde{W}(\vec{\alpha})|$ and average over the measured generalized Wigner value (with an extra phase) at each sampled point. The averaged outcome will converge to the true value as we increase the number of sampled points (and the number of averages at each point as well), which means that our estimator is "unbiased". This is in contrast with the sampling method proposed in Ref. [5, 6] of the SI, where they used $p_{W^2}(\vec{\alpha}) \propto |\tilde{W}(\vec{\alpha})|^2$ and had to discard some sampled points to avoid "pathological behaviour", as you described. Their process could create bias in the form of a systematic difference between true value and estimated value, regardless of the amount of points sampled. To clarify this and elaborate a bit, we have added to the main text: "... without introducing statistical bias, thus ensuring convergence to the true values (see Supplementary Information)."

4. In the caption of Figure 2, it is not clear what "it" is in "to project it onto the real axis".

We have clarified this by replacing "it" with "the generalized Wigner function." Additionally, we have replaced "project" (as you point out in later comments) with "rotate" to make the language more accurate.

5. On page 3, right column, first paragraph, it is mentioned that “For two modes, OLI performs better due to overhead required for the DEMESST approach (see supplementary information)”. Reading the SI, it was not clear to me what is the overhead you are referring to.

The overhead refers to the sampling requirement. Here, we seek to explain why in the 2-mode case, the DEMESST method may need more sampled points or measurements than the OLI method to achieve the same reconstruction fidelity. DEMESST will be more efficient in the regime where $M > 2N$, since (as shown in Eqn. (19) of the SI) we can ignore at least $M - 2N$ unpopulated, irrelevant modes when estimating each element of the density matrix. In this case, the sampling cost for estimating each individual matrix element is independent of M , leading to a lower overhead overall. On the other hand, when $M \leq 2N$, DEMESST may not help for efficiency. DFE for each basis operator $O_{\vec{n}, \vec{n}'}$ only provides a method for unbiased estimation of its expectation value, which may not be the optimal choice for state reconstruction in terms of efficiency.

To help clarify the meaning of the overhead, we have modified the statement in the main text to "For two modes, OLI performs better due to measurement sampling overhead associated with the DEMESST approach (see supplementary information)."

6. On page 4, left column: “These pulses project one or more of the cavity modes to the transmon $|f\rangle$ level. . .”. I get what you are trying to say, but I feel that “projecting” cavity modes to transmon states is not the right way of describing what you are doing. Consider perhaps rephrasing this.

This is a good point, we have rephrased this in the following way to make the statement more accurate: "These pulses transfer the transmon portion of the joint transmon-cavity state of one or more of the cavity modes to the $|f\rangle$ (second excited) level and outside the $|g\rangle - |e\rangle$ qubit space used for the Wigner tomography."

7. A few lines later, again the word “project” is used two more times. Consider perhaps rephrasing these sentences.

We have rephrased these instances similarly to the previous comment, as follows: "...we take the transmon state of the first mode from $|g\rangle$ to $|f\rangle$ and reduce the sampling to the 2-mode $|01\rangle\langle 10|$ operator. This operation allows us to reduce the size of the sampling problem..." We have also corrected the corresponding language in the caption of Figure 2.

We note that we keep the language of projecting in the SI and in some responses to Reviewer 1 when discussing vacuum state projections. We believe that the term is appropriate in those places (with the projection operator defined before Eqn. (19) of the SI). On the other hand, we also agree that it was a confusing term to use in the main text, hence our corrections above. In the main text, we are essentially implementing subsystem tomography, as described in Sec. VI of the SI.

8. I think that details are missing in the discussion about the simulations in the SI. How exactly do you simulate the Wigner tomography? Enough details should be given for the readers to be able to reproduce the calculations. It would be useful to have them explained in the SI.

Thank you for pointing this out, we have rewritten section VIII of the SI to include a much more detailed description of the simulations. There are too many additions to reasonably quote them all here, but a summary of them is: implementing equations and derivations in previous sections to perform measurement sampling for ideal M-mode W states, simulating our generalized Wigner tomography measurements, modeling readout error, and the state reconstruction procedures for DEMESST and OLI.

9. In the simulated fidelities reported in Table 1, were the parameters used in the simulations obtained independently from the Wigner tomography experiments? Or some simulation parameters were fitted to match the experimental results? I think this should be commented.

The simulation results were obtained independently from the tomography experiments. We performed the simulations by modeling the system (using calibration experiments to get the system parameters), then evolving from the ground state to a final state with the same control drives that we used to prepare the states

in the tomography experiment. The final simulated state was then compared to an ideal W state to get the fidelity. We have added the following to the caption of Table 1 to describe this: "The simulated fidelities are obtained by comparing ideal W states to the states obtained from Lindblad master equation simulations with the experimental control drives."

10. What is the transmon readout error? Can you comment on how the readout error, transmon thermal excitations, transmon decay, and transmon dephasing influence the number of shots (10 in your experiment) needed per displacement vector?

The transmon readout fidelity was around 80% for the data collected in the experiment, so the corresponding error was roughly 20%. The error here is defined as the sum of the probabilities of misidentifying $|g\rangle$ as $|e\rangle$ and vice versa. The limiting factor was not the transmon, but rather a low power readout tone that helped minimize readout error caused by cross-Kerr interactions between the readout and multimode cavity. We chose 10 repetitions to balance the theoretical ideal of maximal information from many singleshot measurements, as we now discuss in the beginning of Section VII of the SI, with practical considerations like our imperfect readout fidelity and the time required to run the experiments. In particular, an important consideration was that in our hardware setup, running repetitions of an experiment took much less time than running different displacements (for example, running 5 repetitions for twice as many displacements would take roughly 1.8 times as long as running 10 repetitions with our number of displacements). Thus, we chose 10 averages to be able to complete our measurements in a reasonable amount of time. We also found 10 averages to be sufficient with our readout to properly calibrate the measurements and avoid biased expectation values. We have added the following text to the end of Section VI of the supplement to summarize: "For the data collected in the main text, the transmon readout fidelity was $\approx 80\%$. The error associated with this fidelity is defined as the overlap between $|g\rangle$ and $|e\rangle$, and can be thought of as the sum of the probabilities of misidentifying $|g\rangle$ as $|e\rangle$ and vice versa. The transmon dephasing and relaxation times were not the limiting factor to this fidelity. Instead, it was because we used a relatively low-power readout

tone. The lower power allowed us to decrease the magnitude of the cross-Kerr interaction between the readout cavity and the multimode storage cavity modes, which would cause readout errors by changing the readout frequency and thus its response."

We have also added the following to Sec. VII: "For one, the time required for our hardware to run more displacements was much longer than the time required for more averages. Keeping the total measurement number fixed while running twice as many displacements (half as many averages) would take roughly 1.8 times as long. Furthermore, we wanted to ensure that we obtained accurate measurement results despite our imperfect readout fidelity ($\approx 80\%$). Thus, we chose 10 averages to be a good middle ground for the required measurement time and sufficient to avoid miscalibration and biased expectation values."

We have also added a derivation of why singleshot measurements minimize the variance prior to the above addition in Sec. VII.

11. On page 4, when discussing the number of distinct displacements in DEMESST, it is mentioned that the Total Measurement Number is divided by 10 and by the number of elements. What do you mean by "number of elements"?

The number of elements here refers to the number of distinct basis elements or density matrix elements that are sampled and measured. We have clarified this in the main text as follows: "For the DEMESST method, the number of distinct displacements for each separate basis element is further divided by the number of such basis or density matrix elements."

12. In Section II of the SI, the range of the Kerr nonlinearities given might have typo. According to Table 2, they range from -6 KHz to +9 KHz.

Thank you for this observation, we have corrected the text to the correct range that was provided in the table, so it now reads "from -6 to +9 kHz."

13. In equation (8) in SI, it is not immediately obvious to me why or how the real part is taken. Is it a redundancy to communicate that, despite the complex phase, the quantity you take the real part of is real?

This is true; in general, for physical ρ and θ , the integral indeed has 0 imaginary part and thus is completely real. However, we keep the redundancy of taking the real part to emphasize that aspect. We have added the following sentence after the equation to help communicate this: "For physical ρ and $\vec{\theta}$, explicitly taking the real part is not necessary (as the integral will have 0 imaginary part, but we explicitly take the real part to emphasize that $F(\rho, \sigma)$ is a real quantity."

14. In equation (20) in the SI, what is $p_D(\vec{\alpha}_{\vec{s}})$?

The expression for $p_D(\vec{\alpha}_{\vec{s}})$ is defined earlier, in Eqn. (11). We have added the phrase "where p_D is the probability distribution function defined in Eqn. (11)" after equation (20) as a reminder.

15. In the paragraph just before Section VI of the SI, it says "(see supplementary information)", but we are already there.

Thank you for catching this oversight, we have removed that phrase.

Cristóbal Lledó

REVIEWERS' COMMENTS

Reviewer #1 (Remarks to the Author):

Second report on NCOMMS-23-46562A

The authors have responded to and resolved all of my previous comments and revised the manuscript accordingly, carefully clarifying their claim about complexity scaling of their method. The clarity and scientific soundness of the manuscript have been improved greatly. After the remaining minor comments below have been addressed, I can recommend publication.

Comments:

- Abstract: "for states that can be expressed" I would change this to "for states that can be represented"
- Intro: "method to demystify and reconstruct" What is mysterious about density matrices? I would remove demystify.
- I cannot find what the dashed lines in Fig. 1b are. I assume they are fits. Please give the fitting functions and parameters.
- Table 1: "distance ratio demonstrates exponential improvement" The exponential improvement is not really demonstrated by the data in the table. It would be consistent with almost any increasing function. One could state that it is "is compatible with" exponential improvement, or remove the word exponential.
- "The simulations include error from transmon and cavity decoherence and decay" -> errors
- Fig. 3 caption: "show fidelities versus an ideal" replace "versus" by "to"?
- Supplement: "exponential dependence on N, which is also known as the sign problem." This sentence was added in response to one of my previous questions but I didn't quite follow the argument why the exponential scaling in N is due to a sign problem. Could you give a reference here or refer to a place in the manuscript where this is discussed.
- In the sentence "We simulate using the two methods to reconstruct an ideal" there are some redundant words (2x "simulate" and "stands represents") and a typo...
- "DEMESST scales polynomially with M, while OLI scales exponentially with M." Again, this is not obvious from SFigs. 4 and 5. Please provide a figure where this is backed up quantitatively or phase it in a way that makes clear that this is a theoretical result and not

an observation on the data.

Best wishes,

Martin Gärttner

PS. I somehow think there should be a more efficient way to reconstruct the density matrix once one restricts to a finite N . It seems odd to me to reconstruct all density matrix elements one by one using separate measurements... Need to think more about this.

Reviewer #2 (Remarks to the Author):

My concerns, questions and comments have been appropriately addressed. I think that also the comments of referee #1 have been adequately addressed by the authors. Following the revisions, the manuscript is clearer and more precise and I would therefore like to recommend it for publication.

REVIEWERS' COMMENTS

Reviewer #1 (Remarks to the Author):

Second report on NCOMMS-23-46562A

The authors have responded to and resolved all of my previous comments and revised the manuscript accordingly, carefully clarifying their claim about complexity scaling of their method. The clarity and scientific soundness of the manuscript have been improved greatly. After the remaining minor comments below have been addressed, I can recommend publication.

Comments:

- Abstract: “for states that can be expressed” I would change this to “for states that can be represented”

Thank you for this improved wording, we have implemented this change to the abstract.

- Intro: “method to demystify and reconstruct” What is mysterious about density matrices? I would remove demystify.

We included “demystify” to suggest a pronunciation of the DEMESST method, as we have internally said the acronym like the word “demist.” However, it is true that this could be confusing, and the connection is a bit of a stretch anyway, so following this recommendation we have removed the word demystify.

- I cannot find what the dashed lines in Fig. 1b are. I assume they are fits. Please give the fitting functions and parameters.

Apologies for this omission and thank you for pointing it out, they are indeed fits to exponential and polynomial functions. We have now added to the caption: “Dashed lines indicate fits to exponential and polynomial functions $y = \exp(a + bx)$ with $a = 0.1, b = 3.6$ and $y = \exp(a + b \log(x))$ with $a = 7.9, b = 2.8$, respectively.”

- Table 1: “distance ratio demonstrates exponential improvement” The exponential improvement is not really demonstrated by the data in the table. It would be consistent with almost any increasing function. One could state that it is “is compatible with” exponential improvement, or remove the word exponential.

This makes sense, we have made the suggested correction to state that it is “compatible with exponential improvement.”

- “The simulations include error from transmon and cavity decoherence and decay” -> errors

Looking at it again, we agree that the original statement was too wordy. On the other hand, we do want to summarize what errors we include in the simulation, so following this suggestion, we have modified the statement to read “The simulations include decoherence and state preparation errors...”

- Fig. 3 caption: “show fidelities versus an ideal” replace “versus” by “to”?

Following this recommendation, we have made this change.

- Supplement: “exponential dependence on N , which is also known as the sign problem.” This sentence was added in response to one of my previous questions but I didn’t quite follow the argument why the exponential scaling in N is due to a sign problem. Could you give a reference here or refer to a place in the manuscript where this is discussed.

Apologies for our inaccurate phrasing here. In our case, the exponential scaling in N is not due to the sign problem, but rather the increased dimension of the space corresponding to larger N . Our discussion of the sign problem in response to that question emphasizes: first, from numerical study, the increased oscillation of the sampling distribution with larger N will lead to increased sampling overhead, but the overhead does not increase exponentially; and second, since we focus on the dependence on large mode number M with a bounded total photon number N , we do not consider the sign problem in our problem setup. Since offhandedly mentioning the sign problem is likely to generate more questions and confusion, especially without the related analysis, we have decided to remove that statement. Instead, we more clearly describe the exponential N dependence by replacing it with: “The source of the sampling overhead’s N dependence is the increased subspace dimension and number of O operators. However, the subject of our work is the protocol’s dependence on M .”

- In the sentence “We simulate using the two methods to reconstruct an ideal” there are some redundant words (2x “simulate” and “stands represents”) and a typo...

Thank you for catching this, we have modified the sentence to read as follows:
"We use the two methods to reconstruct an ideal W state $|W\rangle = \frac{1}{\sqrt{M}} \sum_{m=1}^M |1_m\rangle$,
where $|1_m\rangle$ represents the multimode Fock state with a single photon in the m -th
mode."

- "DEMESST scales polynomially with M , while OLI scales exponentially with M ." Again,
this is not obvious from SFigs. 4 and 5. Please provide a figure where this is backed up
quantitatively or phrase it in a way that makes clear that this is a theoretical result and not
an observation on the data.

This is indeed a theoretical result. We have clarified this by rewriting that
sentence as: "Based on the theoretical results from Supplementary Note 5 and
discussed in this Note, DEMESST scales polynomially with M , while OLI scales
exponentially with M ."

Best wishes,

Martin Gärttner

PS. I somehow think there should be a more efficient way to reconstruct the density matrix
once one restricts to a finite N . It seems odd to me to reconstruct all density matrix elements
one by one using separate measurements... Need to think more about this.

Thank you for this suggestion! Reconstructing all the elements one by one with
independent choices of sampling points is indeed less efficient. On the other hand,
it is simpler for our theoretical analysis to achieve a provable Hoeffding's bound
(which requires that each sample should be independent). In the future, we could
try to update the sampled points adaptively based on previous measurement
outcomes, which should improve the resource overhead.

Reviewer #2 (Remarks to the Author):

My concerns, questions and comments have been appropriately addressed. I think that also the comments of referee #1 have been adequately addressed by the authors. Following the revisions, the manuscript is clearer and more precise and I would therefore like to recommend it for publication.

Thank you for supporting our manuscript for publication!